# The impact of adjuvant chemotherapy on survival in mucinous and non-mucinous rectal adenocarcinoma patients after TME surgery

**Karolina Vernmark**[1,2], **Annika Knutsen**[1,2]*, **Per Loftås**[3], **Xiao-Feng Sun**[1,2]

**1** Department of Oncology, Linköping University, Linköping, Sweden, **2** Department of Biomedical and Clinical Sciences, Linköping University, Linköping, Sweden, **3** Department of Surgery, Linköping University, Linköping, Sweden

☯ These authors contributed equally to this work.

* annika.knutsen@regionostergotland.se

**Data Availability Statement:** All relevant data are within the paper and its Supporting information files.

## Abstract

### Introduction

The value of adjuvant chemotherapy for rectal cancer patients is debated and varies in different subgroups. One such subgroup is mucinous adenocarcinoma (MAC), which is more treatment resistant compared to non-mucinous adenocarcinoma (NMAC). To date, mucinous histology is not taken into account when deciding on adjuvant treatment strategy. This is the first study to exclusively include patients with rectal cancer, then separate MAC and NMAC and compare the survival in patients that had or did not have adjuvant chemotherapy.

### Material and methods

The study included retrospective register data from 365 Swedish patients with stage II-IV rectal adenocarcinoma, 56 patients with MAC and 309 patients with NMAC. All patients were considered curative, had surgery with total mesorectal excision in 2004–2013, and were followed up until death or 2021.

### Results

Patients with MAC that had adjuvant chemotherapy had better overall survival (OS, HR 0.42; CI 95%: 0.19–0.93; $p = 0.032$) and a trend towards better cancer-specific survival (CSS, HR 0.41 CI 95%: 0.17–1.03; $p = 0.057$) compared to patients without chemotherapy (HR 0.42; CI 95%: 0.19–0.93; $p = 0.032$). The difference in OS was still significant even after adjusting for sex, age, stage, differentiation, neoadjuvant chemotherapy and preoperative radiotherapy (HR 0.40; CI 95%: 0.17–0.92; $p = 0.031$). There was no such difference in the NMAC patients except in the stage-by-stage subgroup analyses where patients in stage IV had better survival after adjuvant chemotherapy.

**Funding:** The study was supported by grants from Region Östergötland and The foundation of the Departement of Oncology in Linköping, Sweden. The funders had no role in study design, data collection and analysis, decision to publish, or preparation of the manuscript.

**Competing interests:** NO—the authors have declared that no competing interests exist.

## Conclusions

There may be a difference in treatment response to adjuvant chemotherapy between MAC and NMAC patients. Patients with MAC could possibly benefit from adjuvant chemotherapy in stages II-IV. Further studies are however needed to confirm these results.

## Introduction

Every year about 125 000 people in the European Union are diagnosed with rectal cancer (RC). Mucinous adenocarcinoma (MAC), a distinct subtype of RC, is defined by the World Health Organization as an adenocarcinoma in which at least 50% of the cancer tissue are composed of mucin. Two large population-based studies showed that MAC accounted for 18% of all RCs [1, 2].

In CRC, compared to non-mucinous adenocarcinoma (NMAC), MAC has been known to more often present at a later stage [2–4], to be less likely to be resected with negative circumferential margin [5, 6], more often metastasizes to lymph nodes [7, 8] and to be more prone to local recurrence [5, 6] as well as peritoneal carcinomatosis [8, 9]. There are also known differences between colorectal MAC and NMAC in terms of gene expression, such as prevalence of mutations in *p53*, *p21*, and *BRAF* [10, 11].

There are conflicting results regarding the prognosis of CRC patients with MAC compared to NMAC. Most studies report a worse prognosis for rectal MAC [2, 12–14] while others show that from 1999 onwards the survival has been comparable due to the entrance of total mesorectal excision (TME) surgery [6]. Nevertheless, compared to NMAC, MAC has been associated with a poorer response to chemotherapy (CT) and chemo-radiotherapy [15, 16]. A recent study by our group showed that RC patients with NMAC had better survival after short-course radiotherapy compared to long-course radiotherapy, while there was no such difference in the MAC group, indicating that patients with MAC or NMAC may also respond differently to type of preoperative radiotherapy [17]. A study by Souadka *et al* from 2019 including 84 patients with complete pathologic response after neoadjuvant treatment followed by TME, showed that MAC and poorly differentiated tumors were independent factors for worse disease-free survival (DFS) [18].

The use of adjuvant CT to prevent recurrence after surgery *i.e.* adjuvant CT in RC is debated. Some studies show that there is no difference in survival between patients with RC in stage II-III that have adjuvant CT compared to those that do not have adjuvant CT [19]. Other studies show that adjuvant CT improves survival [20, 21] and some yet conclude that there at least is insufficient evidence to say that there is no absolute benefit for adjuvant CT in patients with RC [22]. The inconclusive evidence on the use of adjuvant CT after preoperative chemo-radiotherapy or radiotherapy and surgery for patients with RC is shown by large international differences in treatment guidelines [19]. These differences may be due to the fact that certain, not yet identified, subgroups of patients benefit more or less from adjuvant CT compared to others, but no such subgroups have yet been clearly identified.

Although MAC is different from NMAC in terms of gene expression and histology, the standard adjuvant chemotherapy treatment for colorectal adenocarcinoma is also recommended to rectal MAC patients and no clinical guidelines have been developed specifically for rectal MAC patients. The question is, should mucinous histology affect the recommendation to have adjuvant CT? The aim of this study was to investigate the impact on survival between

RC patients that had adjuvant CT compared to those who did not in the subgroups of MAC and NMAC, respectively.

## Material and methods

### Study design

We performed a retrospective cohort study including initially 528 patients with rectal adeno-carcinoma that had surgery in Region Östergötland, Sweden, between 2004 and 2013. All patients had given their written consent to participate in the study and the study was approved by the Linköping ethical committee. A retrospective medical review from each patient's onco-logical and surgical files was performed. Characteristics of the patients and tumors included sex, age at diagnosis, tumor location, pathologic stage after surgery, histological type, differentiation, lymphovascular invasion, type of surgery, preoperative radiotherapy and/or CT as well as postoperative adjuvant CT. The decision on whether to give the patient preoperative CT and/or adjuvant CT was based on the recommendation from a multidisciplinary team meeting. Patients that were recommended adjuvant CT were planned to receive a total of six months of adjuvant CT. Comorbidities and medications at the time of diagnosis were also registered. Comorbidities were divided into three groups, cardiovascular disease (including any heart disease, hypertonia, stroke and vascular embolism), diabetes (both type I and type II) and pulmonary disease (including asthma and chronic obstructive pulmonary disease). Data including overall survival (OS), cancer-specific survival (CSS), DFS, distant recurrence free survival (DRFS) and local recurrence free survival (LRFS) were collected and considered primary outcome measures. All data was re-confimed by two of the authors (KV and AH), and any uncertainties or disagreements were resolved by consensus. End of follow up was due to either death or end of follow up with was set to 27/05/2021. The mean follow up time was 82 months.

### Analyses

All statistical analyses were conducted with the statistical program Statistica, version 13.5.0.17. The Chi-square ($_x^2$) test was used to examine the relationship between clinicopathological characteristics and treatments. Differences in distance to anal verge, number of adjuvant CT cycles and days from surgery until adjuvant CT between MAC and NMAC patients were calculated with Student's t-test. Survival curves were calculated based on the Kaplan-Meier method. The survival analyses including univariate and multivariate were carried out with the Cox proportional hazard model and presented as hazard ratios (HR) with 95% confidence intervals (CI). Statistical significance was set at $p < 0.05$ for all tests.

## Results

### Clinicopathological characteristics

As shown in Fig 1, among the 528 patients, we excluded 163 patients with stage I disease since these patients did not receive adjuvant CT. Finally, 365 patients were enrolled in the study for further analysis. Among 56 patients with MAC, 19 (34%) received adjuvant CT and 37 did not. Of 309 patients with NMAC, 89 (29%) received adjuvant CT and 220 did not.

   Among the 365 patients included in the study, MAC was found in 56 (15%) of the patients and NMAC in 310 (85%) of the patients. There were no statistically significant differences in sex, age, stage, vascular invasion, perineural growth, distance to anal verge or tumor location between MAC and NMAC patients ($p > 0.05$). There was however a difference in grade of

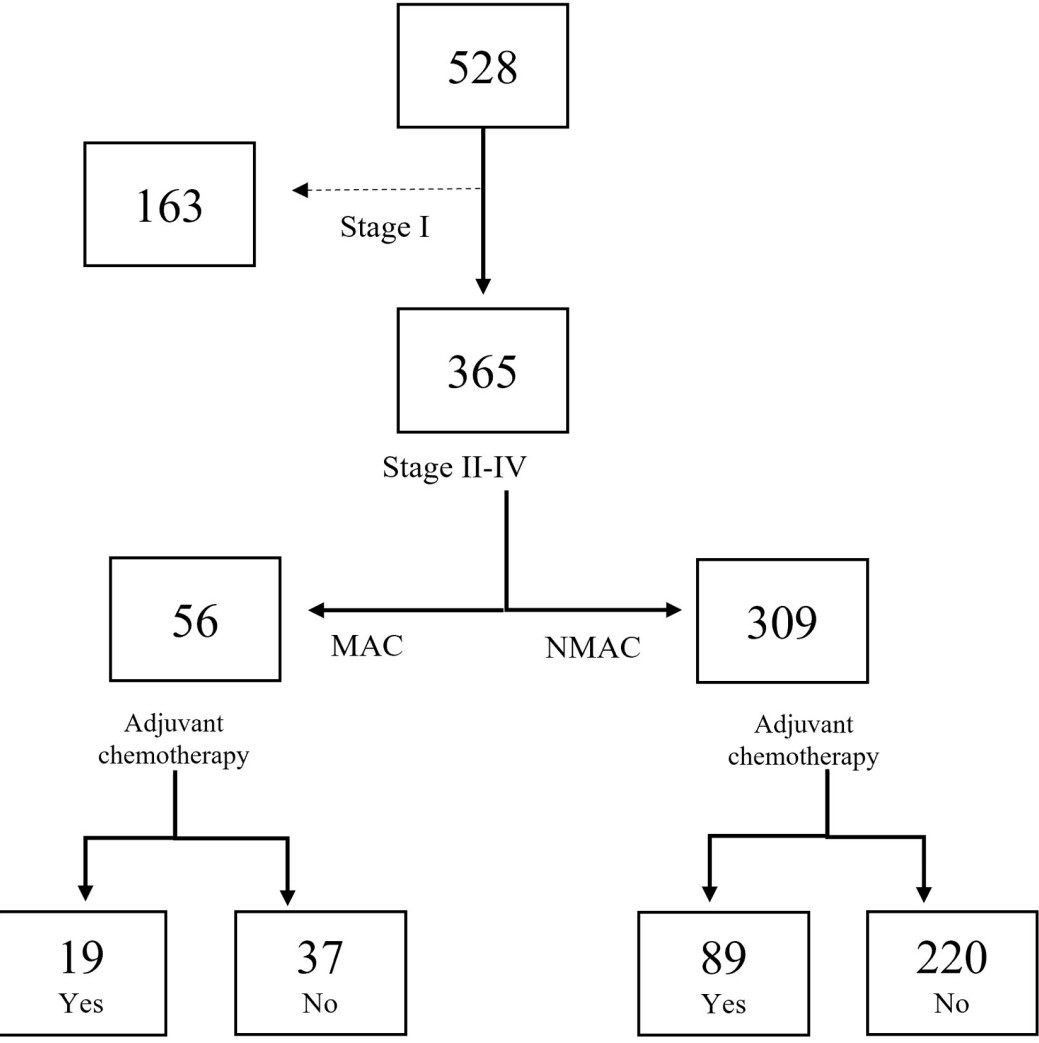

**Fig 1. Flow-chart of rectal cancer patients in the study.**

differentiation where 70% of the MAC was considered poorly differentiated compared to 28% of the NMAC (p <0.001). The results are shown in Table 1.

It was also more common in the MAC group to have inflammatory bowel disease (p < 0.001). In the MAC group three patients (6%) had Chron's disease and two patients (4%) had ulcerative colitis compared to the NMAC group where two patients (<1%) had Chron's disease and one patient (<1%) had ulcerative colitis.

## Preoperative treatments

In total 70 patients had preoperative CT (neoadjuvant CT). Patients with MAC more often had preoperative CT compared to NMAC patients, however the difference did not reach statistical significance (29% *vs.* 17%, $p = 0.052$). Most patients (97%) had 5-FU or a combination of 5-FU and oxaliplatin, and the remaining 3% had 5-FU and irinotecan. There were no differences in the type of preoperative CT ($p = 0.737$).

Table 2 presents the data of treatments and recurrence, where 274 patients (74%) had preoperative radiotherapy, among them, 176 patients (65%) had short-course radiotherapy (25.0

**Table 1. Clinicopathological characteristics of rectal cancer patients.**

| Characteristics | All patients (%) | | | Patients with adjuvant CT (%) | | |
|---|---|---|---|---|---|---|
| | MAC | NMAC | p-values | MAC | NMAC | p-values |
| | n = 56 (15) | n = 309 (85) | | n = 19 (18) | n = 89 (82) | |
| **Sex** | | | 0.683 | | | 0.687 |
| Male | 33 (59) | 173 (56) | | 11 (58) | 47 (53) | |
| Female | 23 (41) | 136 (44) | | 8 (42) | 42 (47) | |
| **Age (years)** | | | 0.211 | | | 0.231 |
| ≤ 69 | 33 (59) | 154 (50) | | 16 (84) | 63 (71) | |
| >69 | 23 (41) | 155 (50) | | 3 (16) | 26 (29) | |
| **Comorbidity** | | | 0.546 | | | 0.176 |
| yes | 28 (50) | 141 (46) | | 10 (53) | 32 (36) | |
| no | 28 (50) | 168 (54) | | 9 (47) | 57 (64) | |
| **Number of comorbidities** | | | 0.469 | | | 0.298 |
| 0 | 28 (50) | 168 (54) | | 9 (47) | 57 (64) | |
| 1 | 24 (43) | 104 (34) | | 8 (42) | 22 (25) | |
| 2 | 4 (7) | 34 (11) | | 2 (11) | 10 (11) | |
| >2 | 0 | 3 (1) | | 0 | 0 | |
| **Comorbiditiy[1]** | | | 0.291 | | | 0.099 |
| Cardiovascular | | | | | | |
| Yes | 28 (50) | 131 (42) | | 10 (53) | 29 (33) | |
| No | 28 (50) | 178 (58) | | 9 (47) | 60 (67) | |
| Pulmonary | | | 0.104 | | | 0.244 |
| Yes | 0 | 14 (5) | | 0 | 6 (7) | |
| No | 56 (100) | 295 (95) | | 19 (100) | 83 (93) | |
| Diabetes | | | 0.320 | | | 0.703 |
| Yes | 4 (7) | 36 (12) | | 2 (11) | 7 (8) | |
| No | 52 (93) | 273 (88) | | 17 (89) | 82 (92) | |
| **Distance to anal verge (cm)** | | | 0.218 | | | 0.909 |
| Mean (cm) | 7.4 | 8.2 | | 7.7 | 7.8 | |
| **Location of tumor** | | | 0.499 | | | 0.232 |
| High (>10 cm) | 5 (9) | 42 (14) | | 1 (5) | 12 (14) | |
| Middle (5–9 cm) | 7 (13) | 28 (9) | | 4 (21) | 8 (9) | |
| Low (< 5 cm) | 43 (78) | 235 (77) | | 14 (74) | 68 (77) | |
| **pTNM stage** | | | 0.467 | | | 0.899 |
| IIA | 14 (25) | 119 (38) | | 3 (16) | 10 (11) | |
| IIB | 2 (4) | 8 (3) | | 1 (5) | 3 (3) | |
| IIIA | 2 (4) | 16 (5) | | 2 (10) | 5 (6) | |
| IIIB | 12 (21) | 55 (18) | | 4 (21) | 27 (30) | |
| IIIC | 15 (27) | 61 (20) | | 6 (32) | 32 (36) | |
| IV | 11 (20) | 50 (16) | | 3 (16) | 12 (14) | |
| **pT-stage** | | | 0.630 | | | 0.787 |
| T1 | 0 | 4 (1) | | 0 | 2 (2) | |
| T2 | 4 (7) | 27 (9) | | 2 (11) | 12 (13) | |
| T3 | 43 (77) | 239 (78) | | 15 (79) | 61 (69) | |
| T4 | 9 (16) | 35 (12) | | 2 (10) | 14 (16) | |
| **pN-stage** | | | 0.272 | | | 0.770 |
| N0 | 20 (36) | 143 (47) | | 5 (26) | 17 (19) | |
| N1 | 16 (29) | 82 (27) | | 7 (37) | 38 (43) | |

*(Continued)*

**Table 1.** (Continued)

| Characteristics | All patients (%) | | | Patients with adjuvant CT (%) | | |
|---|---|---|---|---|---|---|
| | MAC | NMAC | *p*-values | MAC | NMAC | *p*-values |
| | **n = 56 (15)** | **n = 309 (85)** | | **n = 19 (18)** | **n = 89 (82)** | |
| N2 | 19 (35) | 78 (26) | | 7 (37) | 33 (38) | |
| **Vascular invasion** | | | 0.972 | | | 0.285 |
| Yes | 16 (29) | 89 (29) | | 6 (32) | 40 (45) | |
| No | 40 (71) | 220 (71) | | 13 (68) | 49 (55) | |
| **Perineural growth** | | | 0.105 | | | 0.037 |
| Yes | 7 (13) | 68 (22) | | 2 (11) | 31 (35) | |
| No | 49 (88) | 241 (78) | | 17 (89) | 58 (65) | |
| **Differentiation** | | | <0.001 | | | 0.130 |
| Well | 0 | 10 (3) | | 0 | 1 (1) | |
| Moderately | 17 (30) | 211 (68) | | 7 (37) | 54 (61) | |
| Poorly | 39 (70) | 88 (28) | | 12 (63) | 34 (38) | |

CT; chemotherapy, MAC; mucinous adenocarcinoma, NMAC; non-mucinous adenocarcinoma, pTNM; pathologic TNM,

[1]Some patients had several comorbidities

Gy in 5.0 Gy fractions) and 95 patients (35%) had long-course radiotherapy (44.0–50.4 Gy in 1.8–2.0 Gy fractions). Patients generally had capecitabine concomitant with long-course radiotherapy, except for seven patients that had only radiotherapy. None of the patients had postoperative radiotherapy.

Patients with MAC and NMAC were equally likely to have preoperative radiotherapy (73% *vs*. 74%, *p* = 0.848). Patients with MAC however had long-course radiotherapy more often compared to patients with NMAC (46% *vs*. 33%) but the difference was not statistically significant (*p* = 0.100).

## Surgery

All patients had surgery with TME and all surgeries were performed by a consultant surgeon. Most of the patients (n = 217, 61%) had sphincter-sparing surgery. Patients with MAC were less likely to have radical rectal surgery, but the difference did not reach statistical significance (*p* = 0.077). All local recurrences but one occurred before 84 months since the diagnosis. At 24 months after surgery 22 patients (6%) had had local recurrence and by 48 months 32 patients (9%) had had local recurrence.

## Adjuvant CT

Fifty-nine percent of the patients completed the planned six months of treatment. The mean duration of adjuvant CT administered was 4.2 months for all patients, 3.4 months for MAC patients and 4.4 months for NMAC patients. Among the patients, 104 (96%) had 5-fluorouracil (5-FU) alone or a combination of 5-FU and oxaliplatin and four (4%) had 5-FU in combination with irinotecan or irinotecan alone. There were no differences in the types of CT administered in the MAC and NMAC patients (*p* = 0.739) as seen in Table 2.

The mean time from surgery to starting CT was 68 days. There was no difference in time to adjuvant CT from surgery between MAC and NMAC patients (*p* = 0.628). Forty-five patients that had indication for adjuvant CT did not have adjuvant CT. The most common cause was comorbidity and reduced general condition (n = 27, 60%). Other causes were postoperative

**Table 2. Oncological treatment and recurrence in rectal cancer patients.**

| Treatments/recurrence | All patients (%) | | | Patients with adjuvant CT (%) | | |
|---|---|---|---|---|---|---|
| | MAC | NMAC | p-values | MAC | NMAC | p-values |
| | n = 56 (15) | n = 309 (85) | | n = 19 (18) | n = 89 (82) | |
| **Preoperative radiotherapy** | | | 0.848 | | | 0.895 |
| Yes | 41 (73) | 230 (74) | | 16 (84) | 76 (85) | |
| No | 15 (27) | 79 (25) | | 3 (16) | 13 (15) | |
| **Type of radiotherapy** | | | 0.100 | | | 0.160 |
| Short-course radiotherapy | 22 (54) | 154 (67) | | 8 (50) | 52 (68) | |
| Long-course radiotherapy | 19 (46) | 76 (33) | | 8 (50) | 24 (32) | |
| **Preoperative CT** | | | 0.052 | | | 0.189 |
| Yes | 16 (29) | 54 (17) | | 7 (37) | 20 (22) | |
| No | 40 (71) | 255 (83) | | 12 (63) | 69 (78) | |
| **Type of preoperative CT** | | | 0.737 | | | 0.617 |
| 5-FU | 13 (60) | 45 (85) | | 6 (86) | 18 (90) | |
| 5-FU + oxaliplatin | 2 (40) | 6 (11) | | 1 (14) | 1 (5) | |
| 5-FU + irinotecan | 0 | 2 (4) | | 0 | 1 (5) | |
| **Adjuvant CT** | | | 0.439 | | | |
| Yes | 19 (34) | 89 (29) | | 19 (100) | 89 (100) | |
| No | 37 (66) | 220 (71) | | 0 | 0 | |
| **Type of adjuvant CT** | | | 0.739 | | | 0.739 |
| 5-FU | 8 (42) | 44 (49) | | 8 (42) | 44 (49) | |
| 5-FU + oxaliplatin | 10 (53) | 41 (46) | | 10 (53) | 41 (46) | |
| 5-FU + irinotecan | 1 (5) | 2 (2) | | 1 (5) | 2 (2) | |
| Irinotecan | 0 | 2 (2) | | 0 | 2 (2) | |
| **Resection margin rectum** | | | 0.072 | | | 0.686 |
| No tumor cells (R0) | 46 (82) | 279 (90) | | 16 (84) | 78 (88) | |
| Remaining tumor cells (R1) | 10 (18) | 30 (10) | | 3 (16) | 11 (12) | |
| **Local recurrence** | | | 0.799 | | | 0.090 |
| Yes | 5 (9) | 31 (10) | | 0 | 12 (13) | |
| No | 51 (91) | 278 (90) | | 19 (100) | 77 (87) | |
| **Distant recurrence** | | | 0.501 | | | 0.239 |
| Yes | 20 (38) | 96 (33) | | 5 (29) | 39 (45) | |
| No | 33 (62) | 195 (67) | | 12 (71) | 48 (55) | |

MAC, mucinous adenocarcinoma; NMAC, non-mucinous adenocarcinoma; CT, chemotherapy; 5-FU, 5-fluourouracil

complications that delayed treatment to a point where no treatment was meaningful, patient declining treatment and early recurrence. The proportion of patients that had indication for, but did not receive, adjuvant CT were equal between MAC and NMAC patients ($p = 0.953$).

Of the 365 patients included, 70 (19%) had preoperative CT and 108 (30%) had adjuvant CT as shown in Fig 1 and Table 2. Twenty-seven patients (7%) had both preoperative CT and adjuvant CT. Patients with MAC and NMAC were equally likely to have adjuvant CT after having preoperative CT ($p = 0.439$). When including only patients that had adjuvant CT there were no differences between the groups concerning clinicopathological characteristics except for perineural growth that was less common in patients with MAC compared to NMAC (11% vs. 35%, $p = 0.037$, Table 1). There were no differences in the type of CT or radiotherapy treatments as shown in Table 2.

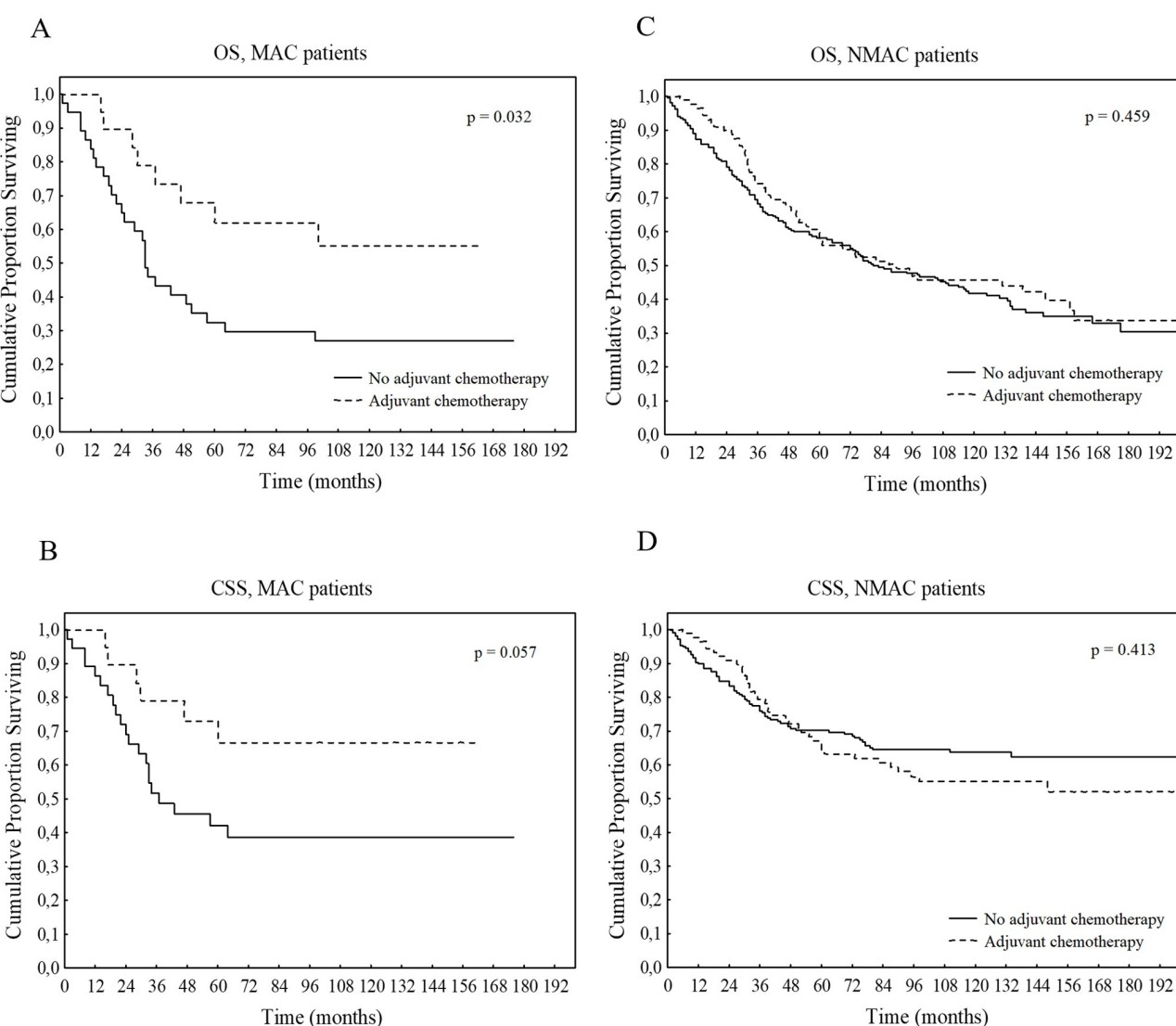

**Fig 2. Survival in patients that had adjuvant CT *vs*. had no adjuvant CT in MAC and NMAC patients.** OS, overall survival; CSS, cancer specific survival.

## Survival

There were no differences in OS or CSS between MAC and NMAC patients after adjusting for stage, or when analyzing MAC and NMAC patient stratified by each tumor stage ($p > 0.05$).

In patients with MAC, there was a significantly better OS (HR 0.42 CI 95%: 0.19–0.93; $p = 0.032$) and a highly trend towards better CSS (HR 0.41 CI 95%: 0.17–1.03; $p = 0.057$) in patients that had received adjuvant CT compared to patients with MAC that had not received adjuvant CT as shown in Fig 2A and 2B. The difference in OS was still significant after adjusting for sex, age, stage, differentiation, preoperative CT and type of preoperative radiotherapy (no radiotherapy/short-course radiotherapy/long-course radiotherapy) (HR 0.40; CI 95%: 0.17–0.92; $p = 0.031$). There was no statistically significant difference in DFS (HR 0.67 CI 95%: 0.28–1.58; $p = 0.357$).

In patients with NMAC, there were no significant differences in OS, (HR 0.89 CI 95%: 0.65–1.22; $p = 0.459$, Fig 2C), CSS (HR 1.18 CI 95%: 0.80–1.74; $p = 0.413$, Fig 2D) or DFS (HR 1.40 CI 95%: 0.97–2.00; $p = 0.067$) even though trending towards a worse DFS with adjuvant CT.

There were no differences in OS, CSS or DFS ($p = 0.389$, $p = 0.491$ and $p = 0.487$, respectively) when comparing patients with MAC and NMAC that had adjuvant CT.

When comparing patients that had received adjuvant CT stage by stage in the MAC and NMAC group, respectively, there were too few patients in stage IIa and IIIa to be able to analyse them separately. Therefore, patients with stage IIa and IIb disease were analysed together, and patients in stages IIIa and IIIb were also analysed together.

There was a significant difference in OS and CSS in patients with MAC that had received adjuvant CT in patients in stages IIIa and IIIb compared to no adjuvant CT (OS: HR 0.14 CI 95%: 0.03–0.74; $p = 0.020$, CSS: HR 0.08 CI 95%: 0.01–0.72; $p = 0.024$).

There was no such difference in MAC patients regarding OS in stage IIa and IIb, (HR 0.60 CI 95%: 0.07–5.0; $p = 0.637$), stage IIIc (HR 0.82 CI 95%: 0.20–3.29; $p = 0.780$) or stage IV (HR 0.34 CI 95%: 0.07–1.66; $p = 0.182$). Analysis of CSS in MAC patients in stage II were not possible due to too small sample size and that 14 out of the 16 patients (87,5%) later died from RC (singular parameter variance matrix). There was no difference between the patients that had or did not have adjuvant CT for patients in stage II regarding prevalence of cancer-related death in the Chi-square test ($p = 0.382$). There were no differences in CSS in stage IIIc (HR 0.96 CI 95%: 0.23–4.04; $p = 0.958$), or stage IV (HR 0.33 CI 95%: 0.07–1.66; $p = 0.182$).

On the other hand, in the NMAC group there was a significant difference in patients with stage IV disease that had received adjuvant CT (OS: stage IV (HR 0.41 CI 95%: 0.19–0.87; $p = 0.021$ and CSS stage IV (HR 0.40 CI 95%: 0.18–0.89; $p = 0.025$).

There was no difference regarding OS in NMAC patients in stages IIa and IIb (HR 0.96 CI 95%: 0.41–2.23, $p = 0.924$), stages IIIa and IIIb (HR 0.85 CI 95%: 0.46–1.55; $p = 0.592$) or in stage IIIc (HR 0.37 CI 95%: 0.36–1.27; $p = 0.230$). There were no differences for NMAC patients regarding CSS in stages IIa and IIb (HR 2.21 CI 95%: 0.74–6.58; $p = 0.153$), stages IIIa and IIIb (HR 1.42 CI 95%: 0.59–3.42; $p = 0.439$), or stage IIIc (HR 0.67 CI 95%: 0.33–1.35; $p = 0.258$).

## Discussion

This is the first study that compares the clinical outcome of MAC and NMAC RC patients separately regarding adjuvant CT. The study showed that RC patients with MAC histology that received adjuvant CT had a significantly better OS compared to patients with MAC that did not have adjuvant CT. The difference was still significant even in the multivariate analysis after adjusting for sex, age, stage, differentiation, preoperative CT and type radiotherapy. This was not the case in patients with NMAC. The differences in survival between MAC and NMAC patients with adjuvant CT could not be explained by differences between the two subgroups regarding relative number of patients that received preoperative CT since the difference was not significant and also did not affect the results in the multivariate analysis. In a stage-by-stage analysis, adjuvant CT was related to better survival in stage IIIA and IIIB in MAC patients. For patients with NMAC there was a significantly better survival in patients with stage IV disease that had adjuvant CT, but not in the other stages. There were no significant differences in the rates of local and distant recurrences in MAC and NMAC patients. At 24 months after surgery, 22 patients (6%) had had local recurrence and, by 48 months, 32 patients (9%) had had local recurrence. The result is comparable to available data, for example, the

Dutch TME trial that had a total of 9% local recurrence including patients that either had or did not have radiotherapy [23].

To date, mucinous histology is not a factor that routinely is taken into account when deciding on whether or not to administrate adjuvant CT for patients with RC, although MAC is different from NMAC in terms of gene expression and histology [10, 11] and there are indications that MAC patients respond differently to treatments [15–17]. There are four randomized controlled trials including patients with RC that, without taking the mucinous histology into account, have compared treatment with adjuvant CT to active expectation after radiotherapy or chemoradiotherapy and surgery. Bosset et al (2007) [24] included 1101 patients and Sainato et al (2014) [25] included 655 patients, and both studies randomized the patients to either postoperative 5-FU and leucovorin or no further treatment. Both the studies had problems with patients being randomized to adjuvant CT but never receiving it. In the Dutch PROCTOR/SCRIPT trial (2015) 177 patients were randomized to adjuvant 5-FU and leucovorin or observation (PROCTOR) and 292 patients were randomized to capecitabine or observation (SCRIPT), respectively [26]. The Chronicle trial (2014) included 113 patients and compared capecitabine and oxaliplatin to no adjuvant CT [27]. Unfortunately, both the PROCTOR/SCRIPT trial and the Chronicle trial were prematurely closed due to poor accrual. The four studies all showed no benefit for adjuvant CT [28], however none of them took the subgroups of MAC and NMAC into account.

There are retrospective studies on colon cancer and adjuvant CT that have studied the effect of adjuvant CT in the subgroups of MAC and NMAC patients. Hugen et al (2013) included 27 251 patients with CRC and then compared the OS in MAC and NMAC patients with or without adjuvant CT in colon cancer patients exclusively [12]. Fields et al (2019) included 31 041 MAC patients with colon cancer stage II-III and compared OS with or without adjuvant CT [29]. Both studies showed better survival for MAC patients that had adjuvant chemotherapy compared to no chemotherapy, but then again only colon cancer patients were analyzed. Only one previous study has exclusively included RC patients and then analyzed the survival and treatment response for MAC patients in regard to adjuvant CT. In the study by Chand et al [30], 191 RC patients with MAC stage I-III were included and the study showed a better 5-year OS for patients that had adjuvant CT after TME surgery.

In our study, we included not only patients with MAC but also patients with NMAC as a reference. We excluded patients in stage I since these patients in general are not recommended adjuvant CT, but in contrast to most other studies, we included patients with stage IV disease. Studies show that 23–36% of patients in stage IV can be alive 5–10 years after surgery with curative intent (including metastasectomy) [31]. Since some of the patients with stage IV disease have a longer OS and CSS compared to some of the patients with stage III disease, it seemed reasonable to include the stage IV patients in this study, which gives us a better perspective of the group of patients that are treated at our clinic and is truly real-world data.

A common remark in studies on adjuvant CT is that patients that have CT in general have a better general condition. A strength in our study is that we have included a reference group—the NMAC patients. If we assume that patients that have adjuvant CT in general have a better general condition compared to patients who do not have adjuvant CT, this should also be the case in the NMAC patients. Thereby all patients that had adjuvant CT should have better survival, which NMAC patients that had adjuvant CT did not have. Furthermore, there were no differences in comorbidities between MAC and NMAC patients that could explain the lack of better survival for NMAC patients that had adjuvant CT.

There are conflicting guidelines regarding adjuvant CT for RC patients in general [28]. This is partly due to a lack of specificity in existing studies. Patients with RC are often included in the same studies as patients with colon cancer, even though they are two different types of

entities. RC is also, compared to colon cancer, more uncommon and larger studies are therefore not conducted as often as they are in colon cancer. There are also very few studies that compare subgroups of colorectal cancer, and fewer yet that compare MAC and NMAC. MAC is more uncommon than NMAC and the combination of RC and MAC is even rarer. The sample of patients used in this study has the advantage of being a relatively large real-life sample which represent Swedish conditions well, however, the number of MAC patients that had adjuvant CT is still small since MAC was only present in 15% of the patients. The percentage of patients with MAC in our study is also consistent with findings in large population-based studies [1, 2]. Our study suggests that patients with MAC and NMAC may respond differently to adjuvant CT. In future studies, instead of only comparing the survival in RC patients with MAC compared to RC patients with NMAC we suggest that the response to therapy should be compared within these subgroups. A large prospective randomized trial would of course better answer the question on the benefit of adjuvant CT in MAC and NMAC patients, but this does not seem feasible. However, a future larger scale retrospective trial may be able to shed some additional light to the question.

## Conclusions

The present study showed that RC patients with MAC may have better survival after adjuvant CT compared to those without adjuvant CT. These results could possibly indicate that rectal MAC and NMAC patients may need different treatment strategies, however further studies are needed to confirm the results.

## Supporting information

**S1 File. Deidentified patient data file.** Data used in the study analyses.
(PDF)

## Author Contributions

**Conceptualization:** Karolina Vernmark, Annika Knutsen.

**Data curation:** Karolina Vernmark.

**Formal analysis:** Karolina Vernmark.

**Funding acquisition:** Karolina Vernmark.

**Investigation:** Karolina Vernmark, Annika Knutsen, Xiao-Feng Sun.

**Methodology:** Karolina Vernmark, Annika Knutsen, Xiao-Feng Sun.

**Project administration:** Karolina Vernmark, Annika Knutsen, Xiao-Feng Sun.

**Software:** Karolina Vernmark.

**Supervision:** Per Loftås, Xiao-Feng Sun.

**Validation:** Karolina Vernmark, Annika Knutsen, Per Loftås, Xiao-Feng Sun.

**Visualization:** Karolina Vernmark, Xiao-Feng Sun.

**Writing – original draft:** Karolina Vernmark.

**Writing – review & editing:** Annika Knutsen, Per Loftås, Xiao-Feng Sun.

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
