## [Decision Letter · Decision Letter 0]

15 Nov 2022

PONE-D-22-16390The impact of adjuvant chemotherapy on survival in mucinous and non-mucinous rectal adenocarcinoma patients after TME surgeryPLOS ONE

Dear Dr. Holmqvist,

Thank you for submitting your manuscript to PLOS ONE. After careful consideration, we feel that it has merit but does not fully meet PLOS ONE’s publication criteria as it currently stands. Therefore, we invite you to submit a revised version of the manuscript that addresses the points raised during the review process.

We look forward to receiving your revised manuscript.

Kind regards,

Mona Pathak, PhD

Academic Editor

PLOS ONE

https://journals.plos.org/plosone/s/fileid=ba62/PLOSOne_formatting_sample_title_authors_affiliations.pdf.

“This study was supported by grants from Region Östergötland and The foundation of the Department of Oncology in Linköping, Sweden.”

“NO -the authors recieved no specific funding for this work”

4. Please ensure that you include a title page within your main document. You should list all authors and all affiliations as per our author instructions and clearly indicate the corresponding author.

Reviewers' comments:

Reviewer's Responses to Questions

**Comments to the Author**

1. Is the manuscript technically sound, and do the data support the conclusions?

Reviewer #1: Yes

Reviewer #2: Yes

2. Has the statistical analysis been performed appropriately and rigorously? 

Reviewer #1: Yes

Reviewer #2: Yes

3. Have the authors made all data underlying the findings in their manuscript fully available?

Reviewer #1: Yes

Reviewer #2: Yes

4. Is the manuscript presented in an intelligible fashion and written in standard English?

Reviewer #1: Yes

Reviewer #2: Yes

5. Review Comments to the Author

Reviewer #1: First, I would like to thank the editor and the authors to the opportunity to review this well written article.

The authors from Sweden investigated a very interesting question, due to the lack of strength in the recommendation of adjuvant chemotherapy in stage 2 to 4 rectal cancer.

The authors succeed in exploring this research idea, using adequate statistical methods. However they need to correct thein conclusion and formulate them more cautiously.

In the Introduction section: The authors should focus more on the bad prognosis of MAC or undifferentiated rectal tumor adding even in complete response to neoadjuvant CRT, as shown in: 10.1186/s12885-019-6239-3 were in Dworack 4 patients, MAC and undifferentiated tumor were an independent factor of of unpaired disease free survival.

In the Study design:

The figure 1 should be in the results section not in the methods.

Results:

The authors should report more details on the quality of surgery: total or partial mesorectal excision, completeness and radicality of the surgery , median circumferential margins, and information on surgeons and their expertise. All these informations may impact the survival of the patients independently of receiving or not adjuvante treatment.

The rate of 10% of local recurrence after TME and preoperative RCC seems a little bit high. The authors should argue this result and give some explanations in the discussion part.

The consultation section should be written more cautiously in both the abstract and the manuscript:

The present study showed that RC patients with MAC may have better survival after adjuvant CT compared to those without adjuvant CT,

These results could possibly suppose that rectal MAC and NMAC patients may need different treatment strategies. and further comparative studies are needed.

Reviewer #2: Though this is a retrospective study but still gives and important message about role of adjuvant chemotherapy in MAC rectal cancer. Possibly because of small sample size of MAC in this retrospective analysis survival benefit between MAC and NMAC could not be show.

This gives rise to an important hypothesis for future trials where MAC may be treated with little more aggressive chemotherapy

6. PLOS authors have the option to publish the peer review history of their article (what does this mean?). If published, this will include your full peer review and any attached files.

Reviewer #1: **Yes: **Amine Souadka

Reviewer #2: **Yes: **DR ATUL SHARMA

---

## [Author Response · Author response to Decision Letter 0]

9 Jan 2023

Dr. Mona Pathak Dec. 15th 2022

Academic Editor

PLOS ONE

Manuscript (MS) ID: PONE-D-22-16390

Manuscript title: The impact of adjuvant chemotherapy on survival in mucinous and non-mucinous rectal adenocarcinoma patients after TME surgery

Dear Dr. Pathak,

Thank you very much for yours and the reviwers’ comments on our manuscript. We have carefully revised our MS according to the questions and suggestions. The detailed response to each suggestion is provided below. Two manuscripts are submitted, one with tracked changes and another without tracked changes. 

Below, comments and suggestions are italicized, and our responses are marked in blue color for easier recognition.

Comments and suggestions from Academic Editor PLOS ONE, Dr Mona Pathak

1. Please ensure that your manuscript meets PLOS ONE's style requirements, including those for file naming. The PLOS ONE style templates can be found at https://journals.plos.org/plosone/s/file?id=wjVg/PLOSOne_formatting_sample_main_body.pdf and https://journals.plos.org/plosone/s/fileid=ba62/PLOSOne_formatting_sample_title_authors_affiliations.pdf

Response: We would like to thank Editor Pathak for the valuable feed back on our manuscript.

We have carefully revised the manuscript according to the PLOS ONE´s style requirements with changes in the formatting due to previously overlooked errors. 

We have also deleted the word “Disease-free survival” (page 5, line 102). Instead, the abbreviation DFS was used since disease-free survival was mentioned earlier (page 3, line 67).

“This study was supported by grants from Region Östergötland and The foundation of the Department of Oncology in Linköping, Sweden.”

“NO -the authors recieved no specific funding for this work”

Response: We have now removed the funding information from the Acknowledgement section and added the corrected funding-related text to the cover letter. We are grateful that you will change the online submission form on our behalf.

Response: We have now added the relevant data in a PDF file.

4. Please ensure that you include a title page within your main document. You should list all authors and all affiliations as per our author instructions and clearly indicate the corresponding author.

Response: We have now added the title page to the main document. All authors and the corresponding affiliations have been listed as per PLOS ONE’s author instructions and the corresponding author is clearly indicated.

Response: We have reviewed the Reference list carefully, there are no errors or retracted articles. The format has been changed to PLOS ONE formatting.

Comments and suggestions from the Reviewer 1, dr Amine Souadka

First, I would like to thank the editor and the authors to the opportunity to review this well written article.

The authors from Sweden investigated a very interesting question, due to the lack of strength in the recommendation of adjuvant chemotherapy in stage 2 to 4 rectal cancer.

The authors succeed in exploring this research idea, using adequate statistical methods. However they need to correct thein conclusion and formulate them more cautiously.

Introduction

R1 The authors should focus more on the bad prognosis of MAC or undifferentiated rectal tumor adding even in complete response to neoadjuvant CRT, as shown in: 10.1186/s12885-019-6239-3 were in Dworack 4 patients, MAC and undifferentiated tumor were an independent factor of of unpaired disease free survival.

Response: We would like to thank Dr. Souadka for the positive evaluation and valuable feed back to our manuscript. We have added the suggested reference to the Introduction section on page 3, lines 64-67.

Study design

R1 The figure 1 should be in the results section not in the methods.

Response: We have moved figure 1 and the corresponding text to the Result section on page 6, lines 118-123.

Results

R1 The authors should report more details on the quality of surgery: total or partial mesorectal excision, completeness and radicality of the surgery , median circumferential margins, and information on surgeons and their expertise. All these informations may impact the survival of the patients independently of receiving or not adjuvante treatment.

Response: All patients had surgery with total mesorectal excision (TME) as described in the section called

Surgery on page 9, line 158. The data on whether or not the surgery was radical is described in table 2, page

8. Unfortunately we do however not have the data on the median circumferential margin available even though

this would be interesting. Information regarding the surgeons expertise has been added on page 9, lines

158-159.

R1 The rate of 10% of local recurrence after TME and preoperative RCC seems a little bit high. The authors should argue this result and give some explanations in the discussion part

Response: Thirty-six patients (10%) had local recurrence after TME in our study. All patients but one had local recurrence before 84 months had passed since the diagnosis. At 24 months after surgery, 22 patients (6%) had had local recurrence and, by 48 months, 32 patients (9%) had had local recurrence. The result was comparable to available data, for example, the Dutch TME trial that had a total of 9% local recurrence including patients that both had and did not have radiotherapy (doi:10.1016/j.ejso.2009.11.011). We have added information about the timing of the recurrences on page 9, lines 161-163 and to the Discussion section, page 14, lines 246-249.

Conclusion

R1 The consultation section should be written more cautiously in both the abstract and the manuscript:

The present study showed that RC patients with MAC may have better survival after adjuvant CT compared to those without adjuvant CT,

These results could possibly suppose that rectal MAC and NMAC patients may need different treatment strategies. and further comparative studies are needed..

Response: We have changed the Conclusion section in the Abstract and in the manuscript on page 15, lines 314-317, according to the suggestions.

Comments and Suggestions from the Reviewer 2, dr Atul Sharma

R2 Though this is a retrospective study but still gives and important message about role of adjuvant chemotherapy in MAC rectal cancer. Possibly because of small sample size of MAC in this retrospective analysis survival benefit between MAC and NMAC could not be show.

This gives rise to an important hypothesis for future trials where MAC may be treated with little more aggressive chemotherapy

Response: We would like to thank Dr. Sharma for the valuable feed back on our manuscript. 

Yours Sincerely, 

Annica Knutsen (former last name Holmqvist), Associated Prof., MD. PhD. 

Department of Oncology, and Department of Biomedical and Clinical Sciences

Linköping University

SE-581 83, Linköping

Sweden

---

## [Editor Report · Decision Letter 1]

10 Feb 2023

The impact of adjuvant chemotherapy on survival in mucinous and non-mucinous rectal adenocarcinoma patients after TME surgery

PONE-D-22-16390R1

Dear Dr. Knutsen,

We’re pleased to inform you that your manuscript has been judged scientifically suitable for publication and will be formally accepted for publication once it meets all outstanding technical requirements.

Kind regards,

Mona Pathak, PhD

Academic Editor

PLOS ONE

---

## [Editor Report · Acceptance letter]

17 Feb 2023

PONE-D-22-16390R1 

The impact of adjuvant chemotherapy on survival in mucinous and non-mucinous rectal adenocarcinoma patients after TME surgery 

Dear Dr. Knutsen:

I'm pleased to inform you that your manuscript has been deemed suitable for publication in PLOS ONE. Congratulations! Your manuscript is now with our production department. 

Kind regards, 

on behalf of

Dr. Mona Pathak 

Academic Editor

PLOS ONE